# Antiangiogenic Phytochemicals Constituent of Diet as Promising Candidates for Chemoprevention of Cancer

**DOI:** 10.3390/antiox11020302

**Published:** 2022-01-31

**Authors:** Ana Dácil Marrero, Ana R. Quesada, Beatriz Martínez-Poveda, Miguel Ángel Medina

**Affiliations:** 1Department of Molecular Biology and Biochemistry, Faculty of Sciences, University of Málaga, Andalucía Tech, and IBIMA, E-29071 Málaga, Spain; anadacil95@uma.es (A.D.M.); quesada@uma.es (A.R.Q.); 2Unidad 741 de CIBER “de Enfermedades Raras”, E-29071 Málaga, Spain; 3CIBER de Enfermedades Cardiovasculares (CIBERCV), E-28019 Madrid, Spain

**Keywords:** cancer prevention, angiogenesis, chemoprevention, phytochemicals, bioactive compounds

## Abstract

Despite the extensive knowledge on cancer nature acquired over the last years, the high incidence of this disease evidences a need for new approaches that complement the clinical intervention of tumors. Interestingly, many types of cancer are closely related to dietary habits associated with the Western lifestyle, such as low fruit and vegetable intake. Recent advances around the old-conceived term of chemoprevention highlight the important role of phytochemicals as good candidates for the prevention or treatment of cancer. The potential to inhibit angiogenesis exhibited by many natural compounds constituent of plant foods makes them especially interesting for their use as chemopreventive agents. Here, we review the antitumoral potential, with a focus on the antiangiogenic effects, of phenolic and polyphenolic compounds, such as quercetin or myricetin; terpenoids, such as ursolic acid or kahweol; and anthraquinones from *Aloe vera*, in different in vitro and in vivo assays, and the available clinical data. Although clinical trials have failed to assess the preventive role of many of these compounds, encouraging preclinical data support the efficacy of phytochemicals constituent of diet in the prevention and treatment of cancer, but a deeper understanding of their mechanisms of action and better designed clinical trials are urgently needed.

## 1. Introduction

### 1.1. Chemoprevention of Cancer

Cancer is the second leading cause of death worldwide, just below cardiovascular diseases [1,2]. Interestingly, around a third of the deaths from cancer are due to the five leading behavioral and dietary risks: high body mass index, low fruit and vegetable intake, lack of physical activity, tobacco use, and alcohol use, features that are characteristic of a Western diet and lifestyle [3]. Two major approaches have been established to decrease cancer incidence and progression, namely, early detection and prevention of cancer [4,5,6,7]. In the last decades, there has been a focus on the chemoprevention of cancer, a concept first defined by Sporn in 1976 as the use of natural, synthetic, or biologic agents to reverse, suppress, or prevent tumor progression [8]. Whereas early detection of cancer is a broadly accepted approach, cancer chemoprevention still remains a matter of debate in the scientific community, mainly due to the heterogenic results obtained from the reported preventive interventions [9]. In this regard, the level at which the specific cancer preventive action is implemented, as well as the targeted population, must be considered [10] (Figure 1): (a) primary prevention, focused on decreasing the incidence of the disease in a broad population, mainly by reducing the exposure to carcinogenic factors; (b) secondary prevention, aimed at reducing the mortality from a particular type of cancer in medium/high-risk populations, with actions focused on early detection and treatment in subclinical stages of the disease; (c) tertiary prevention, targeted to cancer patients in more advanced stages of the disease and devoted to improving the survival rate and the quality of life. Due to the long-term nature of the preventive strategy, the chemopreventive agent should fulfill with some premises named minimal toxicity, low-cost price, and also capability to promote the physiological antitumor responses of the tumor microenvironment [11,12].

### 1.2. Angiogenesis as a Target for Chemoprevention of Cancer

In this review, we focus on angiogenesis, one of the hallmarks of cancer [13,14] (Figure 2). Angiogenesis is the formation of new capillary blood vessels from pre-existing vascular plexus [15]. Despite its physiological role, deregulation of this process is associated with several pathologies, such as rheumatoid arthritis, diabetic retinopathy, and cancer [16,17]. Pathological activation of angiogenesis is a promising target for chemopreventive actions against cancer, since it supports the growth of the primary tumor by supplying it with nutrients and oxygen and promotes the metastatic process by providing an easy way out for tumor cells [15]. In this context, the concept of angioprevention emerges, referring to the prevention of cancer by inhibition and/or stabilization of tumor angiogenesis [11,15]. In line with this, plant-derived compounds are excellent angiopreventive candidates, since toxicities derived from their consumption are low or inexistent, they are easily accessible as they are components of dietary plant foods, and they exhibit pleiotropic biological activity that targets not only tumor cells, but also endothelial or immune cells. Indeed, targeting angiogenesis for cancer prevention using combinations of plant-derived compounds has been proposed as a feasible and promising anticancer approach [18].

In this review, we present some interesting plant-derived compounds contained in the diet, highlighting their potential to inhibit angiogenesis as a main chemopreventive feature. Table 1 summarizes some of the most used assays to study angiogenesis and/or new modulators of angiogenesis.

## 2. Plant-Derived Compounds Constituent of Diet with Antiangiogenic Activity

### 2.1. Phenolic and Polyphenolic Phytochemicals

#### 2.1.1. Flavonols

Flavonols are polyphenolic compounds belonging to the flavonoid subclass, mainly found in plant foods (fruits and vegetables). Different bioactive compounds of this family have been proposed to exert a chemopreventive activity in cancer, including quercetin, myricetin, kaempferol, and fisetin.

##### Quercetin

The bioactive flavonol quercetin (Figure 3a) is mainly found in onions, broccoli, several fruits, such as apples and berries, as well as in olive oil, red wine, and tea. The role of quercetin as a chemopreventive compound is based on its different antitumor reported effects [31], including cell cycle arrest, apoptosis induction or inhibition of antiapoptotic pathways, and inhibition of neutral sphingomyelinase in different cancer models [32,33,34,35,36,37,38]. In addition, the treatment with quercetin boosts the therapeutic effect of some anticancer drugs, revealing a promising role as a potential enhancer compound in chemotherapy [39]. Quercetin has been described as an antiangiogenic compound, targeting tumor-associated angiogenesis, both in vitro and in vivo. In endothelial cells, quercetin affects cell proliferation, migratory potential, tube formation capability, and matrix metalloproteinase (MMP)-2 expression [40] (Figure 4).

Mechanistically, quercetin has been reported to interfere with different regulatory points of angiogenesis. In bovine aortic endothelial cells (BAEC), quercetin inhibited endothelial nitric oxide synthase (eNOS), an enzyme involved in the production of NO and, thus, promoting an increased vascular permeability [45] (Figure 4). It also induces cell cycle arrest in the early M-phase [46], interfering with mitosis and, thus, cellular replication. Cyclooxygenase-2 (COX-2) has been proposed as an additional target for quercetin in angiogenesis inhibition, related to extracellular matrix remodeling [45], thus suggesting a possible preventive use of this compound in COX-2-mediated diseases, such as breast cancer [47]. Additionally, this compound interferes with the mitogen-activated protein kinase (MAPK) pathway by inhibiting extracellular signal-regulated kinase (ERK) phosphorylation and suppresses the vascular endothelial growth factor 2 (VEGFR2)-regulated Ak strain, transforming (AKT)/mammalian target of rapamycin (mTOR)/P70S6K signaling in both endothelial cells [48] and tumor models [49] (Figure 4).

Additional data from in vivo models support the anticancer potential of quercetin [50,51,52], with a noteworthy interest in reversion of chemoresistance [52,53] and in nanoencapsulation of this compound [52,54,55]. Taken altogether, the pleiotropic regulatory actions exerted by quercetin in endothelial cells would explain the potent antiangiogenic effect of this compound. This, together with its direct antitumor effects, supports its potential role in cancer prevention. Provided these are promising experimental results, clinical trials are needed to clearly determine the efficacy of quercetin as a chemopreventive agent in cancer. Currently there is an active clinical trial based on the administration of quercetin as a dietary supplement for squamous cell carcinoma in patients with Fanconi anemia, whose results are expected to be released by 2023 [56].

##### Myricetin

Myricetin (Figure 3b) is a flavonol found in many vegetables, fruits, and herbs. The reported bioactivity of myricetin includes antioxidant, anti-inflammatory, and antitumor effects [57,58,59,60], which support its potential as a chemopreventive compound in cancer. As a matter of fact, it has been reported that myricetin can promote apoptosis in tumor cells via regulation of the B cell lymphoma 2 (Bcl-2) family of proteins, MAPK and Wnt/β-catenin signaling pathways, disruption of reactive oxygen species (ROS) homeostasis, induction of endoplasmic reticulum stress, and DNA damage [61]. The antitumor potential of myricetin has also been related to its capacity to induce cell cycle arrest in cancer cells [62,63]. The particular role of myricetin as an inhibitor of angiogenesis has been studied in different cancer types [64,65]. In a mouse model of skin tumorigenesis, myricetin inhibited UVB-induced angiogenesis, decreasing VEGF, hypoxia-inducible factor 1 (HIF-1), MMP-9, and MMP-13 expression [65] (Figure 4). Additionally, in an assay in which endothelial cells migrate to form spontaneous tubular structures similar to vessels (Table 1), myricetin substantially suppressed tubular structure formation of human umbilical vein endothelial cells (HUVEC) stimulated by VEGF [66]. Myricetin also showed antiangiogenic effects on the in vivo CAM model [64] (Table 1). Interestingly, this compound also inhibited the PI3-K/AKT pathway in both tumor and endothelial cells [64,65,67] (Figure 4). In line with the proapoptotic effect exhibited by myricetin on many tumor cell lines [62,68,69,70,71,72,73,74,75,76,77], this compound has also induced apoptosis in endothelial cells through a reactive oxygen species (ROS)-dependent mechanism [67]. Despite the fact that in vivo studies in rodents support the chemopreventive role of myricetin in different cancer models [78,79,80,81,82,83,84,85], there is no current clinical evidence of its preventive activity in cancer, nor are there any active clinical trials using this compound. Additional in vitro studies suggest a synergistic effect of myricetin with traditional anticancer drugs [86].

##### Other Flavonols

Kaempferol (Figure 3c) and fisetin (Figure 3d) are two bioactive flavonols found abundantly in vegetables, fruits, teas, and herbs, which exhibit interesting properties related to the chemoprevention of cancer. In addition to antioxidant and anti-inflammatory properties [87,88], kaempferol exhibits antitumor activity in different models of cancer [89,90,91,92,93]. The antitumor potential of kaempferol has been greatly assessed in breast cancer in both in vitro and in vivo models. This compound interferes with the pro-cancer activity of 17b-estradiol, bisphenol A, and the xenoestrogen triclosan [94,95,96], suggesting a preventive role of kaempferol in breast cancer. In this same line, kaempferol is also capable of arresting the cell cycle in breast cancer cells, promoting apoptosis via different pathways, and suppressing both migration and invasion [93,97], among other effects, being a potential chemotherapeutic agent. In addition, a preventive role of kaempferol in several types of cancer has been proposed based on its properties to reduce tumor growth, inhibit metastasis, and induce apoptosis of tumoral cells in in vivo models [98,99,100]. Besides the direct effect on cancer cell growth and metastasis, kaempferol exhibits antiangiogenic activity by inhibiting VEGF expression in human ovarian cancer cells, an effect that seems to be mediated by both HIF-dependent and HIF-independent mechanisms [101]. The direct action on VEGF signaling was also reported in HUVEC when kaempferol inhibited angiogenesis by targeting VEGFR-2 and, consequently, downregulated the PI3K/AKT, MEK, and ERK pathways [102] (Figure 4). Additionally, downregulation of COX-2 expression and interference with the NF-κB pathway have been reported for kaempferol [45,65]. Moreover, this compound has been described to induce apoptosis in endothelial cells via the extrinsic pathway in an ROS-mediated p53/ATM/death receptor signaling mechanism [103].

The chemotherapeutic potential of fisetin in cancer and its multi-target mechanism of action leading to cell cycle arrest, apoptosis, and autophagic cell death have been reported in numerous in vitro and in vivo works [31,104,105,106,107]. Interestingly, this antioxidant flavonol shows antiangiogenic activity in vitro and in vivo, inhibiting endothelial survival, proliferation, migration, and tube formation [108,109] (Table 1). Although its inhibitory mechanism in tumoral angiogenesis is not completely described, fisetin has been reported to inhibit the expression of several molecules implicated in angiogenesis modulation and degradation of the extracellular matrix in both prostate carcinoma and lung carcinoma cell lines, as is the case of VEGF, eNOS, iNOS, MMP-2, and MMP-9 [108,109,110] (Figure 4). Interestingly, fisetin was proposed as an anticancer agent in combination with the traditional cisplatin to increase its cytotoxicity in human embryonal carcinoma NT2/D1 cells as a promising co-therapy, but, despite some rather interesting in vitro results (it was possible to reduce cisplatin concentration fourfold) [111], this approach has not reached the clinical practice yet.

#### 2.1.2. Isoflavones

Consumption of soy food has been traditionally related to Asian culture, although, over the past 25 years, the majority of Western countries have markedly increased the use of soy products in their diet [112]. Despite the controversy regarding the effect of soy-derived compounds on cancer [113], research evidence suggests a beneficial role of soy food in health, especially associated with a decreased risk of developing certain types of cancer, such as prostate and breast cancer [114,115,116]. Isoflavones are polyphenolic compounds from the isoflavones subclass, included among the soybeans components and characterized for their capability to modulate estrogen receptor (ER) activity (phytoestrogenic activity) [117].

Genistein (Figure 3e) is one of the major isoflavones in soybeans, together with daidzein, biochanin A, and others. This compound has been shown to inhibit cancer growth in vitro and in vivo [118,119,120,121,122], affecting cell cycle progression, apoptosis, angiogenesis, cell invasion, and metastasis. Mechanistically, genistein inhibits AKT, NF-κB, MMPs, and Bax/Bcl-2 signaling pathways, among others [119,123] (Figure 4). Interestingly, recent evidence supports the ability of genistein, together with other phytochemicals, such as curcumin, epigallocatechin-3-gallate (EGCG), or resveratrol, to eliminate cancer stem cell populations [124]. In addition, genistein is also a potent inhibitor of angiogenesis, and several mechanisms have been reported, both in endothelial and cancer cells, to be responsible for this activity, including the inhibition of protein tyrosine kinase activity and MAPK activation in VEGF-stimulated endothelial cells [118,123,125] and the inhibition of HIF-1 in pancreatic carcinoma cells [118,125,126] (Figure 4). Genistein downregulates the expression of VEGF, PDGF, and MMPs (Figure 4) and upregulates, on the contrary, the expression of endogenous angiogenesis inhibitors, such as plasminogen activator inhibitor-1, endostatin, angiostatin, and thrombospondin-1 [118,123,127,128,129].

Clinical trials have been performed to evaluate the possible preventive and therapeutic effect, two different applications [113], of genistein in different types of cancer. In a phase 2 clinical trial with patients with localized prostate cancer, short-term genistein intervention modulated the expression of several biomarkers related to prostate cancer prediction and progression, supporting a preventive role of genistein in this type of cancer [130]. In contrast, intervention with genistein in phase 2 clinical trials with bladder, pancreatic, and breast cancer patients failed to determine the chemopreventive activity of this compound, although the possibility of combining genistein treatment with other agents is a promising approach to exert a clinically effective preventive action [131,132,133]. In fact, in September 2019, genistein was shown to be safe and tolerable in combination with FOLFOX (a type of chemotherapy) or FOLFOX-Bevacizumab (chemotherapy plus an antiangiogenic therapy) to treat patients with metastatic colorectal cancer in a phase 1/2 pilot study, the reason why the investigators crave a larger clinical trial to accurately prove the efficacy of this treatment [134].

#### 2.1.3. Green Tea Polyphenols

Green tea polyphenols have shown interesting chemopreventive and antiangiogenic properties. Indeed, regular consumption of green tea is associated with a reduced risk of some types of cancer [135,136]. One of these polyphenols is epigallocatechin-3-gallate (EGCG, belonging to the catechins family; Figure 3f), a potent antioxidant compound with pleiotropic activities. EGCG has been described as an anti-inflammatory, antiangiogenic, and antitumor compound that targets immune, endothelial, and tumor cells [137,138,139,140,141]. Two of its direct effects on tumor cells are modulation of cancer cell response to chemotherapy and induction of apoptosis [138]. The potent antiangiogenic activity of this compound can be mechanistically explained by its capability to regulate the VEGF/VEGFR axis at different levels: EGCG inhibits VEGF expression in cells, interferes with the binding of this ligand to VEGFR2, and reduces the activation of the receptor in endothelial cells [140,141,142] (Figure 4). In addition, other mechanisms have been proposed for the antiangiogenic effect of EGCG, pointing to the modulation of HIF-1α and NF-κB pathways [143,144] (Figure 4) and to the inhibition of endoglin/Smad1 signaling, an alternative proangiogenic pathway that can be activated in tumors to overcome the anti-VEGF therapy [145]. Interestingly, the chemopreventive potential of EGCG in oncologic diseases is reinforced by experimental results supporting the specificity of this compound on tumor-associated endothelial cells and endothelial progenitor cells in contrast to normal endothelial cells [146]. All the observed effects of this compound in cancer cells and tumor-associated processes make it feasible to point to EGCG as the main compound responsible for the beneficial activity of green tea in cancer prevention. Nevertheless, despite the promising chemopreventive properties of EGCG, it is not clear if this compound is able to prevent the initiation or progression of cancer, since clinical trials are still inconclusive or pending results. Differences in bioavailability, dosages, and potential side effects of EGCG are some of the aspects that make it difficult to come to a clear conclusion about its preventive potential [147,148,149], and further clinical studies are needed to verify this issue. Notably, a clinical trial performed in 2016 proved the safety of a decaffeinated catechin mixture containing 200 mg EGCG to be further tested for prostate cancer prevention or other indications [150].

#### 2.1.4. Other Phenolic and Polyphenolic Compounds

The list of plant-derived compounds with antitumor and antiangiogenic effects that are potentially active in cancer prevention is rather extensive and still increasing in length. In addition to the above-mentioned polyphenolic phytochemicals, there are some others that deserve special attention due to their high rates of consumption in many countries, especially in those with adherence to the Mediterranean diet [12]. This is the case for resveratrol (a polyphenolic compound found in grapes and red wine; Figure 3g) and hydroxytyrosol (a phenolic compound found in virgin olive oil; Figure 3h) [151,152]. 

The potential benefits of resveratrol in tumor prevention could be mediated by suppression of the activation of NF-κB [153] and the AKT/STAT3 signaling pathway [154]. Furthermore, modulation of the VEGF/VEGFR2 pathway through several axes (HIF-1α and GSK3β/β-catenin/TCF) seems to be the main mechanism of action of resveratrol in angiogenesis inhibition [155,156,157,158,159,160,161]. Despite the amount of available information from in vitro studies that indicate the potential therapeutic effects of resveratrol in cancer, the in vivo studies remain controversial [162,163,164,165] and the number of clinical trials focused on assessing its use as a chemopreventive agent are still very limited and showed nonconclusive results [166,167]. Additional efforts are needed to gain a deeper understanding of the targets and mechanisms of action of resveratrol and to clinically determine its health benefits.

Hydroxytyrosol, a bioactive compound constituent of virgin olive oil, exhibits multiple health-related properties [152]. Interestingly, our group found that this compound is a multitarget inhibitor of angiogenesis in vitro and in vivo [168,169], highlighting the potential of hydroxytyrosol and its synthetic derivatives [170,171] in cancer prevention.

### 2.2. Terpenes

Terpenes are a large group of natural compounds derived from the isoprene structural unit, and they are classified attending to the number of isoprene units contained in the molecule (monoterpenes, diterpenes, triterpenes…). Many of these compounds are interesting in terms of biomedical research due to the pleiotropic biological activities exhibited, in addition to their easy availability, as they are components of many edible plants and fruits [172]. Natural terpenes, such as ursolic acid, celastrol, carnosol, carnosic acid, and kahweol, are good candidates for chemoprevention of cancer, showing promising antitumor and antiangiogenic activities.

#### 2.2.1. Ursolic Acid

Ursolic acid (Figure 3i) is a pentacyclic triterpene present in the leaves of several plants used as herbal infusions or spices (rosemary, lemon balm, vervain, oregano, sage, thymus) and fruits (apples, cranberries) [173,174]. This phytochemical exhibits pleiotropic biological activity [175,176], which probably derives from its capability to inhibit the NF-κB pathway and STAT3 activation, both in vitro and in vivo [177,178,179,180,181,182,183]. In addition, data from several groups, including ours, showed that ursolic acid has antiangiogenic properties, inhibiting key steps of angiogenesis in vitro [184,185,186]. However, a study reported that ursolic acid increased the expression of angiogenesis modulators, such as VEGF and FGF-2, in endothelial cells by a mechanism that implied the PI3K–AKT signaling pathway [187] (Figure 4). The antiangiogenic potential of ursolic acid has been confirmed in vivo in different murine animal models [180,184,188]. In addition, the bioavailability of ursolic acid after oral administration in mice, and toxicity, pharmacokinetics, and pharmacodynamics of liposomal ursolic acid in humans have been studied, showing promising results for its use in chemoprevention [189], although clinical trials are required to explore its efficacy and applicability to clinical practice.

#### 2.2.2. Kahweol

Kahweol (Figure 3j) and cafestol are antioxidant diterpenes present in coffee beans and unfiltered coffee beverages. Their pharmacological interest is derived from their anti-inflammatory and antitumor properties [190,191]. Among the antitumor effects, these diterpenes have shown antiangiogenic activity by inhibiting angiogenesis in vitro, ex vivo (mouse aortic ring assay), and in vivo (CAM and zebrafish intersegmental vessel models) (Table 1) through downregulation of the VEGFR2 levels and modulation of the VEGF pathway [192,193] (Figure 4), and targeting inflammatory molecules, such as COX-2 and monocyte chemoattractant protein-1 (MCP-1), in endothelial cells [194,195]. The interest in proposing kahweol and cafestol as potential chemopreventive compounds in cancer mainly relies on the high rate of consumption of coffee all around the globe.

#### 2.2.3. Other Terpenoids

Celastrol (Figure 3k), carnosol (Figure 3l), and carnosic acid (Figure 3m) represent good candidates for chemoprevention of cancer. In addition to its antioxidant, anti-inflammatory, and antitumor properties [196,197], the pentacyclic triterpenoid celastrol, present in the traditional Chinese herb *Tripterygium wilfordii* Hook, has been described as an antiangiogenic compound, modulating several pathways, including IKK/NF-κB (IκB kinase/NF-kB) pathway and HIF-1α [197,198,199,200,201] (Figure 4). Carnosol and carnosic acid, active constituents of the rosemary herb, are also rather interesting diterpenes for cancer prevention, mainly due to their capability to inhibit angiogenesis in vitro and in vivo by a mechanism that implies the induction of apoptosis in endothelial cells [202,203].

### 2.3. Anthraquinones

Anthraquinones are an important group of natural and synthetic compounds derived from the molecule of anthraquinone. Important biologically active plant-contained anthraquinones have been described, showing interesting pharmacological activities, such as antitumor and anti-inflammatory effects [204]. Natural anthraquinones, such as emodin and aloe-emodin, are attractive compounds due to their potential chemopreventive properties in cancer [205].

#### 2.3.1. Emodin

Emodin (Figure 3n) is a hydroxyanthraquinone found in molds, lichens, and edible plants, such as rhubarb. This compound has shown anticancer activity in different human cancer cell lines in vitro and in vivo [205,206,207,208], inducing apoptosis and inhibiting epithelial to mesenchymal transition (EMT) in colon cancer [209,210,211] and epithelial ovarian cancer cells [212]. Several reports indicated the antiangiogenic potential of emodin in the context of cancer [213]. This compound inhibited proliferation in endothelial cells by cell cycle arrest in the G2/M phase and induction of apoptosis, together with the inhibition of MMPs and VEGFR2 signaling. Furthermore, emodin has been described as a tyrosine kinase inhibitor, downregulating the phosphorylation of ERK 1/2, which contributes to its antiangiogenic effect [214,215,216] (Figure 4). In vivo experiments show the efficiency of emodin to inhibit angiogenesis in pancreatic cancer, coursing with the regulation of the TGF-β/Smad pathway and angiogenesis-associated microRNAs miR-20b, miR-155, and miR-210 [217,218]. In breast cancer, emodin suppressed angiogenesis, downregulating Runx2 transcriptional activity [219]. Despite the promising results and the pleiotropic antitumor and antiangiogenic mechanism of action, the chemopreventive potential of emodin in cancer remains to be elucidated. Interestingly, a recent clinical study showed that administration of emodin attenuated the adverse effects caused by tamoxifen treatment via cyclin D1 and pERK upregulation in ER-positive breast cancer cell lines [220].

#### 2.3.2. Aloe-Emodin

Aloe-emodin (Figure 3o), present in *Aloe vera* leaves, is a hydroxyanthraquinone with antitumor activity [205,221]. Our group reported the antiangiogenic potential of this compound based on its ability to inhibit urokinase secretion, as well as tubule formation of endothelial cells [222] (Table 1). The antiangiogenic role of aloe-emodin has been further supported by the observation that this compound downregulates MMP-2/9, RhoB, and VEGF in colon cancer cells, interfering with NF-κB binding activity [223] (Figure 4). Interestingly, a report that evaluated the antiangiogenic activity of aloe-emodin in the context of hypoxia-induced retinal angiogenesis showed that this compound interferes with the HIF-1α/VEGF pathway in vitro and in vivo [224] (Figure 4). However, the effect of aloe-emodin regarding angiogenesis should be further studied and clarified, since studies on in vivo models have related the wound-healing capacity of *Aloe vera* to proangiogenic effects [225].

## 3. Concluding Remarks and Future Challenges

The use of nontoxic plant-derived natural compounds as chemopreventive agents in cancer has become an attractive approach in the last years, since it represents a low-cost, easily accessible, and broad-spectrum alternative to conventional drugs. Additionally, the high number of bioactive compounds contained in plant foods that display antitumor and antiangiogenic effects in experimental models should prompt us to undoubtedly assess the feasibility of their use in cancer prevention.

While broad experimental evidence obtained for these compounds clearly points to their capability to target events involved in cancer initiation and progress, their application to cancer prevention is still far away from being clinically established. For many of these compounds, clinical trials are still very limited, contradictory, inconclusive, or even inexistent, making it extremely difficult to clarify the level of efficacy of these compounds in cancer prevention, if any, as well as their preventive role on the population and the suitable regimens needed to achieve the desired health benefits. In this context, a significant effort is needed to improve the design of clinical trials, paying special attention to the target population, as well as to the pharmacokinetics and pharmacodynamics of the compounds. In terms of cancer treatment, however, the use of these compounds in combination with typical therapeutic agents comes as a rather interesting approach, as it would allow the reduction in the aggressiveness and side effects caused by these drugs, while also improving their effects.

Undoubtedly, a deeper understanding of the mechanism of action of the compounds in the context of the hallmarks of cancer is necessary, mainly attending to the pleiotropic character of their reported biological actions. In line with this, it is worth encouraging basic, translational, and clinical research to better integrate the available information to improve the outcome of clinical trials, and eventually determine the health benefits and efficacy of plant-derived bioactive compounds in the chemoprevention of cancer.

## Figures and Tables

**Figure 1 antioxidants-11-00302-f001:**
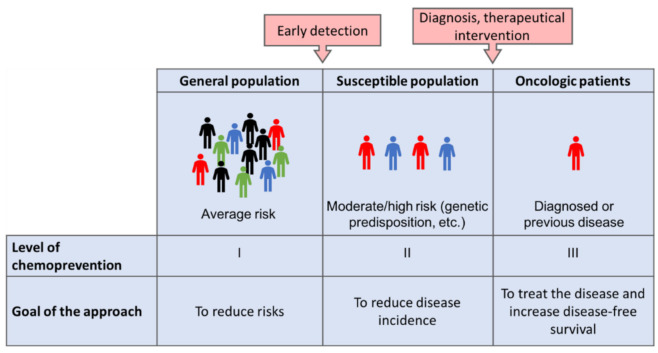
Levels of chemoprevention in cancer. The figure shows the three levels (I, II, III) at which a chemopreventive strategy can be implemented, and the different goals and targeted populations. (Level I of chemoprevention targets the general population, with the aim of reducing risks of developing cancer. Level II of chemoprevention faces the reduction in the incidence of the disease in moderate/high-risk patients who have, for instance, a genetic predisposition to develop a certain type of cancer. Level III of chemoprevention focuses on patients who are diagnosed or have recovered from previous cancer, with the objective of treating the disease and increasing disease-free survival). Modified from [12].

**Figure 2 antioxidants-11-00302-f002:**
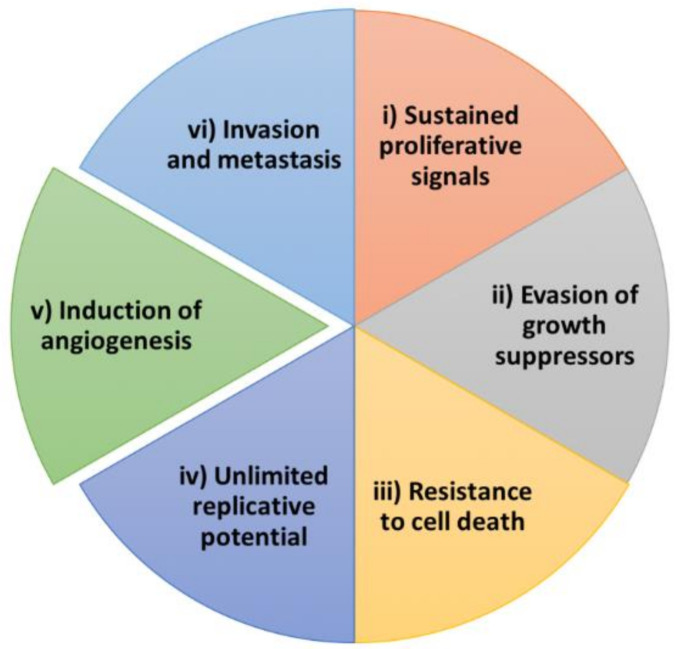
Hallmarks of cancer. The six hallmarks of cancer described by Hanahan and Weinberg in 2000 [13]. The issue was revisited in 2011 [14], when four new hallmarks (not shown in the figure) were added to the list. In 2022, the list of hallmarks has increased up to 14 [19].

**Figure 3 antioxidants-11-00302-f003:**
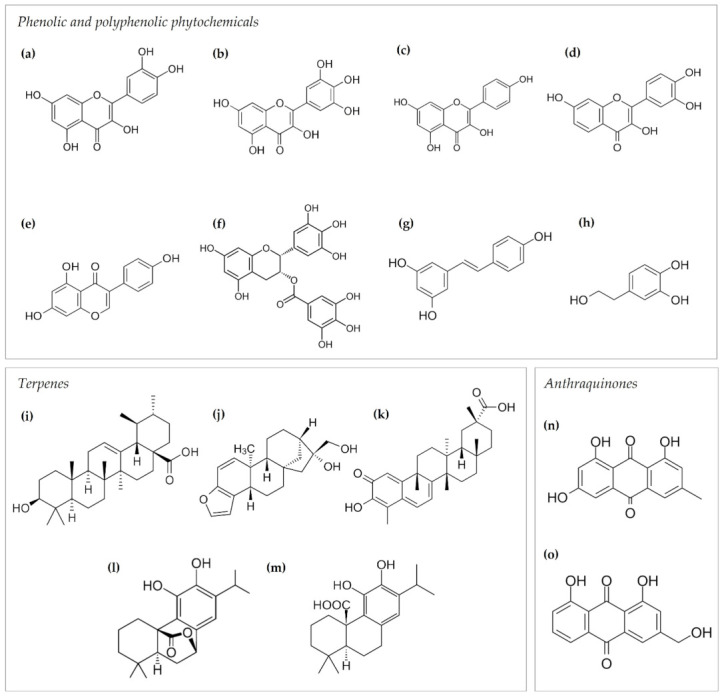
Chemical structures of the phenolic and polyphenolic phytochemicals (**a**) quercetin, (**b**) myricetin, (**c**) kaempferol, (**d**) fisetin, (**e**) genistein, (**f**) epigallocatechin-3-gallate, (**g**) resveratrol, and (**h**) hydroxytyrosol; the terpenes (**i**) ursolic acid, (**j**) kahweol, (**k**) celastrol, (**l**) carnosol, and (**m**) carnosic acid; and the anthraquinones (**n**) emodin and (**o**) aloe-emodin.

**Figure 4 antioxidants-11-00302-f004:**
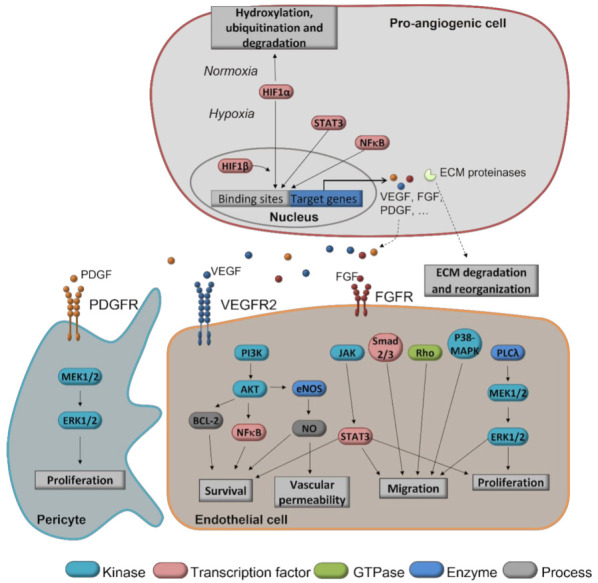
Summary of some of the most important signaling pathways, molecules, and processes involved in the angiogenic process. Proangiogenic cells (i.e., cancer cells) are a source of proangiogenic molecules, such as growth factors, for example, VEGF; cytokines; and ECM-degrading proteases, such as MMPs, that can be received by endothelial cells and pericytes, activating their differentiation, proliferation, survival, and migration to form new vessels. The proteases released by proangiogenic cells degrade the ECM to allow the migration of endothelial cells, releasing as well proangiogenic factors from ECM elements [41,42,43,44]. HIF-1α/β: hypoxia-inducible factor 1-alpha/beta; STAT3: signal transducer and activator of transcription 3; NF-κB: nuclear factor kappa-light-chain-enhancer of activated B cells; VEGF(R): vascular endothelial growth factor (receptor); FGF(R): fibroblast growth factor (receptor); PDGF(R): platelet-derived growth factor (receptor); MEK1/2: mitogen-activated protein kinase kinase 1/2; ERK1/2: extracellular signal-regulated kinase 1/2; PI3K: phosphoinositide 3 kinase; eNOS: endothelial nitric oxide synthase; NO: nitric oxide; BCL-2: B-cell leukemia/lymphoma 2; JAK: Janus tyrosine kinase; Smad 2/3: mothers against decapentaplegic homolog 2; P38-MAPK: 38-kDa mitogen-activated protein kinase; PLCλ: phospholipase C-λ.

**Table 1 antioxidants-11-00302-t001:** Some of the most used assays to study angiogenesis and/or new modulators of angiogenesis in vitro, ex vivo, and in vivo. EC: endothelial cells; MTT: 3-(4,5-dimethylthiazol-2-yl)-2,5-diphenyltetrazolium bromide; ECM: extracellular matrix; MMP: matrix metalloproteinase; CAM: chorioallantoic membrane [20].

	Assay	Description	References
In vitro	Proliferation/survival (MTT)	Angiogenesis depends directly on the active proliferation of EC. The MTT assay measures cellular viability based on the reduction of soluble MTT(3-(4,5-dimethylthiazol-2-yl)-2,5-diphenyltetrazolium bromide) to a blue-colored tetrazolium salt by mitochondrial reductases, followed by detection in a spectrophotometer.	[20,21,22]
	Tubular-like structure formation	Reorganization of EC to create tubular-like networks resembling blood vessel formation. Matrigel, a solubilized murine extract of basal lamina, is used as a substrate.	[21]
	Migration/Invasion	Migration throughout the ECM and its degradation are the main capacities for EC to form new vessels. The migration and invasion fluorometric assays are based on the disposal of fluorescence-labelled EC on a transwell system that allows the movement of EC towards chemotactic stimuli contained in the media in the well. Specific to the invasion assay, a thin layer of Matrigel is placed over the filter of the transwell. Afterward, the light emitted by fluorescence-labeled cells is measured.	[21]
	Zymography (detection of gelatinase and caseinase activities)	Matrix metalloproteinases (MMPs) and serin proteases degrade the ECM to ease the rearrangement of EC to form new vessels. Finding inhibitors of these enzymes can limit the angiogenic process. The detection of their activity involves the electrophoresis of the secreted protease enzymes through discontinuous nondenaturing polyacrylamide gels containing enzyme’s substrate (either type III gelatin or β-casein). Staining the gel with a protein dye allows the detection of the proteolytic activity as clear bands of lysis against a stained background.	[21,23]
Ex vivo	Mouse aortic ring	Culture of transversal sections of rat aorta placed on Matrigel induces the outgrowth of the ring at day 7–14 that can be observed in the microscope.	[24]
In vivo	Chorioallantoic membrane	The strong blood irrigation of the chorioallantoic membrane makes it an economic, feasible model to study angiogenesis. Methylcellulose discs containing the studied compounds can be directly placed on the membrane to test their effect on embryonic angiogenesis.	[20,25,26]
	Zebrafish intersegmental vessel models	Zebrafish is currently a model system for the study of angiogenesis during its embryonic development. After being treated with the compounds to test, the circulatory system of the embryos (transparent) can be easily observed by microscopy or binocular scopes, and abnormalities in the development can be assessed. The detection can be improved by in situ hybridizations, among other approaches.	[27,28,29,30]

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
