# Peer review of "Antiangiogenic Phytochemicals Constituent of Diet as Promising Candidates for Chemoprevention of Cancer"

_antioxidants, 2022, doi:10.3390/antiox11020302_

Round 1

Reviewer 1 Report

The manuscript provides a review on the interesting topic of antiangiogenic potential of natural compounds in chemotherapy. I have just few remarks that in my opinion would improve the quality of the text:

1. The table 1 provides some of the most used assays to study angiogenesis and/or new modulators of angiogenesis. Thus I would recommend to place it before where the general topic is described instead to cite it only in relation to the particular compound – fisetin. The same hold true for the figure 5 that provides a summary of some of the most important signaling pathways, molecules and processes involved in the angiogenic process.

2. I would suggest to uniform the classification of the bioactive compounds described in the text – in some cases the group is natural product (soy isoflavones; green tea polyphenols) in other – chemical class (terpenes; flavonols).

3. I would suggest to merge figures 3 and 4 in order to provide comparative representation of the chemical structures of the compounds that are discussed.

Author Response

First of all, we would like to thank Reviewer 1 for all his/her suggestions and positive criticism, as well as for the overall positive evaluation of our review.

  1. Regarding the table 1, taking into account your suggestion, we have replaced it at the end of section 1.
  2. Thank you for your suggestion. We have re-ordered the contents and subsections of section 2 using as first order criterion the chemical class.
  3. As per the reviewer 1 suggested, we have merged previous figures 3 and 4 in a new, reordered Figure 3.

Reviewer 2 Report

Specific comments:

Since the Authors focused on angiogenesis in their review, it should be reflected in the title of the manuscript.

The Abstract is too general and does not cover the content of the manuscript.

Why is Mediterranean diet among the key words? It is not a topic of the manuscript, and it was mentioned in the text just one time.

More details concerning the best sources of all (not some) basic presented natural compounds will be desirable (in Table?)

The data on chemopreventive study on animal models were scanty (e.g., in case of myricetin and ursolic acid).

In Figure 5 the definition of pericytes (i.e., a type of cells that cover and support the blood vessels) should be omitted.

All abbreviations applied in the text should be explained in the place, they appear for the first time. Some abbreviations were not explained (see e.g., MAPK - line 107, VEGFR2 - line 108), some were explained in the inappropriate place (e.g., EGCG appeared in line 196 but explained in 222), and some were explained two times (e.g, ROS lines 124 and 136). Abbreviations applied in the Figures should be explained as well (e.g, below the caption - see Figures 3 and 5).

The statement “Funding: Our experimental work is supported by grants….” is strange in review manuscript.

Author Response

First of all, we would like to thank Reviewer 2 for all his/her suggestions and positive criticism, as well as for the overall positive evaluation of our review.

  1. Regarding the title, to fulfil your suggestion we have decided to change it. The new title is : Anti-angiogenic phytochemicals constituent of diet as promising candidates for chemoprevention of cancer.
  2. We have completely rewritten the abstract trying to better cover the content of the manuscript.
  3. We have detached “Mediterranean diet” from the list of keywords.
  4. We have added details on the sources of the mentioned natural compounds.
  5. We have added more (available) data regarding studies on animal models.
  6. The new Figure 4 legend has omitted the definition of pericytes.
  7. OK. We have done so.
  8. We have detached the words “Our experimental work is”.

The different data added to the amended version of our manuscript has extended our list of cited references from 205 to 225. We have also corrected some minor errors.
